



# The benefit of using an ensemble of seasonal streamflow forecasts in water allocation decisions

Alexander Kaune[1,2,3], Faysal Chowdhury[1], Micha Werner[1,4], James Bennett[5,6]

[1]Water Science and Engineering Department, IHE Delft Institute for Water Education, Delft, the Netherlands
5 [2]Wageningen Institute for Environment and Climate Research, Wageningen University & Research
[3]FutureWater, Costerweg 1V, 6702 AA Wageningen, the Netherlands
[4]Division of Inland Water Systems, Deltares, Delft, the Netherlands
[5]CSIRO Land & Water, Clayton, Victoria, Australia
[6]Institute for Marine and Antarctic Studies, University of Tasmania, Battery Point, Tasmania, Australia

*Correspondence to*: Alexander Kaune (a.kaune@futurewater.nl)

**Abstract.** The area to be cropped in irrigation districts needs to be planned according to the allocated water, which in turn is a function of the available water resource. Initially conservative estimates of future (in) flows in rivers and reservoirs may lead to unnecessary reduction of the water allocated. Though water allocations may be revised as the season progresses, 15 inconsistency in allocation is undesirable to farmers as they may then not be able to use that water, leading to an opportunity cost in agricultural production. We assess the benefit of using reservoir inflow estimates derived from seasonal forecast datasets to improve water allocation decisions. A decision model is developed to emulate the feedback loop between simulated reservoir storage and water allocations to irrigated crops, and is evaluated using inflow forecasts generated with the Forecast Guided Stochastic Scenarios (FoGSS) model, a 12-month ensemble streamflow forecasting system. Two forcings are used to generate 20 the forecasts: ESP (historical rainfall) and POAMA (calibrated rainfall forecasts from the POAMA climate prediction system). We evaluate the approach in the Murrumbidgee basin in Australia, comparing water allocations obtained with an expected reservoir inflow from FoGSS against the allocations obtained with the currently used conservative estimate based on climatology, as well as against allocations obtained using observed inflows (perfect information). The inconsistency in allocated water is evaluated by determining the total changes in allocated water made every 15 days from the initial allocation 25 at the start of the water year to the end of the irrigation season, including both downward and upward revisions of allocations. Results show that the inconsistency due to upward revisions in allocated water is lower when using the forecast datasets (POAMA and ESP) compared to the conservative inflow estimates (reference) which is beneficial to the planning of cropping areas by farmers. Overconfidence can, however, lead to an increase in undesirable downward revisions. This is more evident for dry years than for wet years. Over the 28 years for which allocation decisions are evaluated, we find that the accuracy of 30 the available water estimates using the forecast ensemble improves progressively during the water year; especially one and a half months before the start of the cropping season in November. This is significant as it provides farmers additional time to make key decision on planting.

*Keywords*: Hydrological information, water scarcity, medium term planning decisions, irrigated agriculture.





## 1 Introduction

Allocating water is the process of sharing the available water among claimants over a period of time (Hellegers and Leflaive, 2015; Le Quesne et al., 2007). Basin authorities are responsible to allocate water among different users; including agriculture, cities, industry and the environment. The available water in rivers and reservoirs, and the demand placed on it, may vary over
time and space due to climate variability, climate change and population growth. Hence, allocating the available water poses a challenge for decision makers, especially in increasing drought and water scarcity conditions.

Basin authorities can allocate water following a demand-based approach. This consists of reviewing the expected water demand in the basin at the start of each water year and allocating the required volume (Linés et al., 2018). Over the water year the initial allocation may, however, be revised depending on how the availability of water in reservoirs and from upstream
catchments evolves. In other basins the water availability is initially reviewed before allocating water. Allocation of water to meet the entitlements of the license holders is based on the estimate of the available water, which is made using the observed stock in the reservoirs at the time of making the decision, as well as the expected inflow.

In Australia, the water allocation process is governed by clear water policy and regulations at basin level, such as defined in the Murray-Darling Basin Plan (Australian government, 2008). The Murray-Darling river system is highly regulated,
especially in the basins in New South Wales (Ribbons, 2009). Predicting the inflow into the reservoirs is key to adequately allocate water, especially for allocation to irrigated agriculture. However, conservatively low estimates of the expected inflow based on climatology are currently used at the beginning of the water year to estimate the available water for the coming season. As the irrigation season progresses, the estimate of available water may be revised. Given the conservative initial estimate the revision is typically upwards, with consequent upward revisions in the water allocated during the year. For most
years the allocated water is set too low at the beginning and is then progressively increased until the 'real' estimate of available water is reached and the 'real' allocation can be established. Ideally, water users would like to know their 'real' water allocation at the beginning of the water year. This is especially so for irrigators that depend on accurate and timely water allocation to choose which crop to plant and to decide on the area to be cropped, allowing them to maximise the benefit of the water that their license entitles them to.

The use of seasonal forecasts of reservoir inflows may be beneficial to support water allocation decisions through providing better and earlier estimates of the available water. The potential for farmers to benefit from seasonal forecasts will, however, depend on i) how well the climate can be predicted, ii) how much this information helps in the actual decision process and iii) how much it contributes to reducing negative impacts (Hansen, 2002). Most studies focus on evaluating the benefit of forecasts by determining the skill, often based on forecast results, observed data and a benchmark prediction (Pappenberger et al., 2015).
Though this provides insight to the (relative) quality of forecast, it may say little of the benefit to users through improved decisions. The use of seasonal forecasts to support decisions has been addressed in several settings. Winsemius et al. (2014) assess the predictability of meteorological indicators in a changing climate and show how skilful forecasts can support rain-fed agriculture. Shukla et al. (2014) developed and implemented a seasonal agricultural drought forecast system for East Africa





which shows to perform well in drought years. Crochemore et al. (2016) assessed the performance of seasonal streamflow forecasts to a reservoir with standard indicators of forecast skill such as reliability, sharpness, and accuracy. Anghileri et al. (2016) evaluate the performance of Ensemble Streamflow Prediction (ESP) forecasts based on climatology and perfect forecasts. Turner et al. (2017), in addition to the usual forecast skill assessment, included the performance gain in reservoir

operation, benchmarking penalty costs when using forecast against using perfect and actual information. Boucher et al. (2012) applied an ensemble streamflow forecast for determining its value in supporting hydropower generation. However, the potential enhancement to water allocation decisions in irrigated agriculture that are informed by seasonal forecasts has been little studied. A complete assessment of the added value of seasonal forecasts can allow basin authorities to explore the opportunities seasonal forecasts provide to improve their operational decisions, and reduce potential losses to agricultural by

improving water allocation estimates.

In this study, we develop and test a water allocation framework to assess the value of using seasonal forecasts of inflow into reservoirs in the regulated Murrumbidgee basin in Australia. We hypothesise that water allocation decisions can be improved when informed by skilful and reliable seasonal forecasts of reservoir inflows. The inconsistency in water allocation decisions during the water year is introduced as a measure of the value of using the forecast in the allocation process under the assumption

that if water allocated to farmers changes little during the season that they can then make maximum benefit of their allocation. If the inconsistency in water allocation decisions using the forecast is lower than when using the currently used conservative estimate of water availability, then there is value in using the forecast. In addition to the water allocation estimated at the start of the water year, key decision dates for the cropping season are evaluated to determine the benefit to the farmers in supporting the decisions they need to make as the season progresses.

## 2 Methods

### 2.1 Water allocation process in the Murrumbidgee basin, Australia

The regulated Murrumbidgee River basin (84000 km²) was selected to evaluate the benefit of using forecasted reservoir inflow (Figure 1). In this basin, two major water storages, the Burrinjuck and Blowering reservoirs, provide the required resource for the water allocation process. The New South Wales Office of Water announces the water allocation for different users in the

basin starting July 1st of each year based on the available water in storage, the expected reservoir inflows for the next 12 months, and water requirements downstream. The inflow to the Blowering reservoir depends on both the discharge release from an upstream hydropower system (Snowy Hydro Scheme) and the natural runoff, while the inflow to the Burrinjuck reservoir depends mainly on natural runoff. Currently, a conservative estimate of the total inflow to the storages of $2.33 \times 10^8 \ m^3/year$ (based on climatology) is used to determine the expected reservoir inflow for the next 12 months.

In the Murrumbidgee basin, the water year runs from July 1st through to June 30th of the next calendar year, while the cropping season for annual (irrigated) crops is between November 1st and the end of February. Each water user (e.g. irrigation, urban and environment) holds a water entitlement (water share), which is a license to extract water from the basin. Depending on the





available water, the basin authority announces a fraction of the total volume of that entitlement that will be met in a year, which is defined as the water allocation (Green, 2011). Water is allocated to different water users according to established priorities. In water abundant conditions each user gets their agreed full entitlement. When water shortage occurs, the highest priority is to satisfy human water consumption and the lowest priority is to satisfy the irrigation demand of annual crops. This is reflected

in the type of license users pay for, with irrigators holding a High Security license having a higher priority to water than irrigators holding a General Security license. Normally, irrigators growing annual crops hold a General Security license, though it is the annual crops that require the highest volume of water. Once the water has been allocated, users can decide how they would like to employ the resources available to them. A maximum of 30% of the volume entitlement can be carried over from one year to another. This provides flexibility to the water users as they can hold a certain volume of their allocation in the

storages and make it available for the next year.

During the water year the basin authority may revise the initial water allocation and announce a new allocation for each user depending on the then available water storage volume and the estimate of the remaining inflow to the end of the water year. In the currently established regulations the water allocation cannot be decreased, but only maintained or increased, unless exceptional circumstances dictate. This process generates expectations among the irrigators about the amount of water volume

they are going to get, especially for the General Security (GS) license holders. To illustrate the process, Figure 2 shows the recorded water allocation decisions made in the 2016-2017 water year. On July 1$^{st}$ 2016, the initial water allocation for General Security (GS) license holders was established at 40%. Two weeks later, the water allocation was increased from 40% to 52%. Successive revisions over the next four months resulted in a final water allocation of 100% at the start of the cropping season (November), which was subsequently maintained until the end of the season.

Certainly, irrigators are aware that the allocated water can be higher in the announcements made closer to the start of the cropping season. This means that farmers may well act on the expectation that the allocation is typically revised upwards, taking the risk of a certain water volume finally being allocated. We argue that this risk affects decision making among the farmers and their ability to correctly plan the area to be cropped. Farmers more averse to risk may tend towards cropping according to the conservative water allocation, leading to potential losses (opportunity cost), while less risk averse farmers

may face losses in yield if the final allocation they expect is not met. Ideally, water users would like to be informed about the true water allocation as early as possible to better plan their activities. This means that the ideal allocation scenario is when the water allocation is set at the start, and then remains constant during the water year.

## 2.2 Data and information

Data and information from the Murrumbidgee basin was collected from the online repositories of the New South Wales Office

of Water and the Australian Bureau of Meteorology (BoM). Actual water allocation for the different users (e.g. General Security) and the observed daily inflow into the reservoirs were obtained for the period 2011 to 2016 (www.water.nsw.gov.au). The current water allocation policy was introduced in 2011, with allocations prior to that date following a different policy. For the 1982-2009 period, actual daily inflow into the two reservoirs was back-calculated using observed daily outflows and





observed storage. Discharge data from gauging station 410008 (Murrumbidgee river downstream of Burrinjuck dam) and gauging station 410073 (Tumut river at Oddys bridge) were used to obtain the daily outflow from Burrinjuck and Blowering reservoirs, respectively. Daily reservoir storage volumes were obtained for each reservoir from station 410131 (Murrumbidgee river at Burrinjuck dam – storage gauge) and station 410102 (Tumut river at Blowering dam – storage gauge) (see Figure 1).

The forecasted datasets for determining the expected reservoir inflow were obtained from an experimental streamflow forecasting system called Forecast Guided Stochastic Scenarios (FoGSS), available for the time period 1982-2009 (Bennett et al., 2018). Two different datasets from FoGSS were used. One dataset is generated from precipitation and sea-surface temperature (SST) predictions from the POAMA M2.4 seasonal climate forecasting system (Hudson et al., 2013; Marshall et al., 2014). The other dataset is generated with historical precipitation, similar to Extended Streamflow Prediction (ESP) (Day,

1985). A more detailed description of FoGSS is provided in section 2.3.

To determine the expected reservoir inflows, FoGSS forecasts for the 10304 km² upstream basin of the Burrinjuck reservoir are considered. Inflows to the Blowering reservoir are dominated by releases from the upstream Snowy Hydro Scheme, which includes various inter-basin transfers and unknown operating rules. Inflows into the Blowering reservoir from the Snowy Hydro scheme are therefore taken as observed and not subject to the forecast, with the observed releases from Snowy Hydro

discharge obtained through back-calculating from observed outflows gauged at Oddy's Bridge just downstream of the dam and observed storage data of the Blowering reservoir (1982-2009). Forecasts for the Goobarragandra River at Lacmalac gauge, a main tributary of the Tumut River between the Blowering Reservoir and its confluence with the main Murrumbidgee, with an influence basin area of 668 km² (Figure 1) are considered. This means that the water available for allocation is determined as the available storage in the Burrinjuck and Blowering reservoirs; the observed inflows to the Blowering reservoir from the

Snowy Mountain Scheme; and the FoGSS forecast flows for the Goobarragandra River and for the Murrumbidgee River upstream of the Burrinjuck reservoir.

### 2.3 Forecast Guided Stochastic Scenarios (FoGSS)

Forecast Guided Stochastic Scenarios (FoGSS) is an experimental ensemble streamflow forecasting system, which has been developed and tested for the Australian continent (Bennett et al., 2016, 2017; Turner et al., 2017). FoGSS produces forecasts

in the form of monthly time series for a 12-month forecast horizon. As forecast skill declines with lead time, FoGSS is designed to nudge forecasts towards climatology. To produce streamflow forecasts, FoGSS forces a monthly hydrological model with reliable ensemble rainfall forecasts. Hydrological uncertainty is then quantified and propagated through the forecast with a staged error model, ensuring reliable ensembles. FoGSS combines skill available from rainfall forecasts and initial hydrological conditions to produce streamflow forecasts that are at least as skilful as climatology. In this study we make use

of two different rainfall forcings to generate FoGSS forecasts, termed ESP+ and POAMA. ESP+ forecasts are generated with an ensemble of historical rainfall sequences to force the hydrological model. This is similar to well-established ESP methods (e.g. Day (1985)), with the difference that streamflow forecasts are also processed with the FoGSS error model to produce reliable ensembles (hence denoted as ESP+). The FoGSS error model allows the generation of large ensembles, and each





'ESP+' forecast is made up of 1000 ensemble members. The POAMA forecasts are generated with post-processed SST and rainfall forecasts from the POAMA M2.4 seasonal climate prediction system. These forecasts are processed with the method of calibration, bridging and merging (Schepen and Wang, 2014) to correct biases, remove noise and ensure reliable rainfall forecast ensembles. POAMA forecasts combine skill from seasonal climate prediction with skill from initial hydrological

conditions, while ESP+ forecasts rely on skill only from initial hydrological conditions. As with the ESP+ forecasts, the POAMA-driven inflow forecasts each have an ensemble of 1000 members. Various configurations of FoGSS have been trialled by Bennett et al. (2017): they found that the GR2M monthly hydrological model (Mouelhi et al., 2006) and the use of a Bayesian prior in the error model generally produced the best performance, and this is the configuration used in this paper. Full details of FoGSS can be obtained in Bennett et al. (2016, 2017).

**2.4 Developing the water allocation decision model**

The water allocation decision making process developed in the Murrumbidgee basin consists of a feedback loop between the simulated available water resource and the emulated water allocation decision for the different users. Figure 3 schematically shows the decision model to establish the resource available to the different users. All time dependent variables are from day t to the end of the water season. Users include, in order of priority; Environmental Water (EW=$0.60 \: x \: 10^8 \: m^3/year$), Towns

(TD=$0.85 \: x \: 10^8 \: m^3/year$), High Security (HS=$3.60 \: x \: 10^8 \: m^3/year$), Irrigation Conveyance (IC=$3.76 \: x \: 10^8 \: m^3/year$), and General Security (GS=$18.9 \: x \: 10^8 \: m^3/year$).

The available water is determined at a daily time step and the water allocation decision is emulated for selected announcement dates. The available water is determined considering the storage volume in the reservoirs; expected reservoir inflow (1982-2009 average $31.5 \: x \: 10^8 \: m^3/year$); storage reserves, and water losses. The expected inflow into reservoirs is the input

variable, which feeds into the established water balance to determine the available water for allocation. Water allocation decisions are emulated for the different users from which the water allocation for GS is derived. As water is released from the reservoirs due to the water allocation process, a new water availability estimate is determined for the next time step.

**2.4.1 Determining the available water for water allocation**

The available water for allocation on the first day ($t = 1$) of the water years is defined as (Equation 1):

$$AW(1) = S + I_a - L_a - R \hspace{6cm} (1)$$

where $S$ is the water storage in the reservoirs using the observed storage for the first day, $I_a$ is the expected annual inflow into the reservoirs, $L_a$ is the expected annual water loss ($7.71 \: x \: 10^8 \: m^3/year$), $R$ is the annual reserve of water storage with a fixed value of $R$=$1.52 \: x \: 10^8 \: m^3/year$ including the storage reserve, dead storage, and unusable inflow).

The expected annual inflow into the reservoirs is based on the regulated inflow from the Snowy Hydro scheme, and a natural

inflow. In the current operating policy, the natural inflow into both reservoirs is established at $2.33 \: x \: 10^8 \: m^3/year$, which is a conservative inflow estimate, corresponding to the 3% non-exceedance probability of the annual inflow distribution from





climatological data. In our approach we replace this fixed inflow with the forecasted inflows derived from the FoGSS ensemble. The actual inflow volume considered in the allocation decision is established using selected non-exceedence percentiles of the summed 12 month inflow prediction of FoGSS.

The available water for allocation, $AW$ starting on the second day ($t \geq 2$) includes the water used due to allocation $U$ (Equation 2):

$$AW(t) = S(t) + I_e(t) - L_e(t) - R + U_c(t) \tag{2}$$

where $S(t)$ is the simulated storage obtained with $S' = S(t-1) + \Delta S$ and the capacity of the reservoirs $C$. If $S' < C$ then $S(t) = S'$, but if $S' \geq C$ then $S(t) = C$ with a spill $sp = S' - C$. The storage change is defined as $\Delta S = I_o(t-1) - U_d(t-1) - L_d(t-1)$ where $I_o$ is the observed daily inflow, $L_d$ is the daily expected loss determined from the expected annual loss distributed equally for each day, and $U_d$ is the daily expected allocation for the different water users. The daily expected allocation for General Security, Irrigation Conveyance and High Security and Towns is determined with a daily release ratio multiplied by the water allocation for each user. For General Security, Irrigation Conveyance and High Security the daily release ratio is based on the monthly irrigation requirements (Table 1), while for towns the water allocation is distributed equally for each day. The water allocation for each user is obtained based on the daily available water from the water balance and the allocation rules explained in detail in the next paragraph. $I_e(t)$ is the expected inflow for the remaining days of the water year defined as $I_e(t) = I_e(t-1) - I_d(t-1)$, where $I_d$ is the daily expected inflow determined from an established daily inflow fraction of the average annual observed inflow on day $t-1$. $L_e(t)$ is the expected loss for the remaining days of the water year defined as $L_e(t) = L_e(t-1) - L_d(t-1)$, where $L_d$ is the daily expected loss to the previous day. $U_c(t)$ is the cumulative water use due to allocation defined as $U_c(t) = U_c(t-1) + U_d(t-1)$, where $U_d$ is the daily expected allocation for the different water users obtained with the daily release ratio and the emulated water allocation decision. The water allocation decision is emulated each day, but the final allocated water is presented only for announcement dates; starting on July 1st and then for each 15 days to the end of the season. In order to emulate the water allocation decision the framework includes the available water, the agreed entitlement for each water user and the priority rules. The volume is established stepwise for each user following the priority of water use. The allocated water for Towns, stock, domestic and basic right (TD) was set at 100% of the entitlement. Only if the difference between the available water for allocation and the environmental water is lower than $0.85 \times 10^8 \ m^3/year$ (the entitlement of TD), then the allocated water for TD is the difference of AW and EW. The procedure for allocating water to High Security (HS) is presented in Figure 3. It includes the use of the available water AW and the allocated water for TD and EW to obtain three possible outcomes for the allocated water for HS. Depending on availability, water allocation for HS can be 100% or 95% of the entitlement, or the difference of the AW, TD and EW. The allocated water for General Security is determined by using the remaining water volume left after the High Security procedure, subtracting the amount lost to Irrigation Conveyance ($3.76 \times 10^8 \ m^3/year$). If that difference is lower than the General Security entitlement, then the allocated water for GS is lower than 100%.





The decision model is tested against the recorded allocation decisions for the years from 2011 to 2016 as the current water allocation policy was introduced in 2004 (Horne, 2016).

### 2.4.2 Evaluating water allocation decisions

To evaluate the water allocation decisions made during the water year a metric to quantify the inconsistency in allocated water is introduced. The inconsistency ($I$) can occur due to either upward or downward revisions of the allocated water volume during the water year. An upward revision is when the allocated water at time step $t$ ($WA_t$) is larger than the allocated water at time step $t-1$ ($WA_{t-1}$). Hence, the inconsistency in allocated water due to upward revisions ($I^+$) is the sum of the difference between allocated water at time step $t$ ($WA_t$) and the allocated water at time step $t-1$ ($WA_{t-1}$) with that condition:

$$\forall\, WA_t > WA_{t-1} \Rightarrow I^+ = \sum_{t=1}^n WA_t - WA_{t-1} \tag{3}$$

A downward revision occurs when the allocated water at time step $t$ ($WA_t$) is lower than the allocated water at time step $t-1$ ($WA_{t-1}$), and the inconsistency in allocated water due to downward revisions ($I^-$) is thus the sum of the absolute difference between allocated water at time step $t$ ($WA_t$) and the allocated water at time step $t-1$ ($WA_{t-1}$) with that condition:

$$\forall\, WA_t < WA_{t-1} \Rightarrow I^- = \sum_{t=1}^n |WA_t - WA_{t-1}| \tag{4}$$

A constant water allocation from the beginning until the end of the water year implies zero inconsistency. This would imply
that the expected inflow estimates are perfect, and the total water allocation is correctly determined at the start of the season. This water allocation $WA_p$ was derived by applying the observed inflows in the decision model. We separate dry and wet years according to the allocated water obtained with the observed inflow. Years where the allocated water is equal to 100% of the entitlement are considered as wet years, while years where the allocated water is lower than 100% of the full entitlement are considered dry years. Average water allocation decisions for dry and wet years were obtained at each time step. A second
metric, the Root Mean Square Difference($RMSD$) was used to evaluate the allocated water obtained with the expected inflows $WA_i$ against the allocated water obtained with the observed inflows (perfect information) $WA_p$ at each time step $t$ for selected years $y$ (dry or wet years) (Equation 5).

$$RMSD = \sqrt{\frac{\sum_{y=1}^m (WA_i - WA_p)_y^2}{m}} \tag{5}$$

## 3 Results

### 3.1 Emulating historical water allocation decisions

A preliminary calibration of the allocation framework was developed for the 2011-2016 water years using the conservative inflow estimate ($2.33\, x\, 10^8\, m^3/year$), and the recorded allocation decisions. The main calibration parameter is the allocation use reduction factor, which determines the percentage of the water allocated to them that users decide to use, with the remainder



being reserved for carry-over to the next year. In reality, this factor varies between users as well as between years, and may be influenced by a variety of factors, including many that do not depend on water availability. We simplify this by considering a bulk allocation use reduction factor across all General Security users and also consider this to be equal across years. The allocation use reduction factor derived for the 2011-2016 period is then assumed to also hold for evaluating the FoGSS datasets

for the 1982-2009 water years, for which recorded allocation decisions are not available as a different policy for the allocation of water to the different users was then in place. The allocation use reduction factor was established as 78%, for which similar simulated and actual carry over volumes are obtained, as well as the simulated storage in the reservoirs at the end of each water year (itself a function of the carry-over volume which remains in the reservoir). Figure 4 shows the simulated water storage in the reservoirs, as well as the observed and simulated carry-over volumes. For the years 2011 to 2016, the emulated water

allocation decisions for General Security (GS) are shown in Figure 5, and compared to the actual allocation decisions recorded in those years. Two simulations are shown. In the first, the initial storage condition of the reservoirs (day 1 of the 2011 water year) was set equal to the actual water storage followed by an open loop simulation as in equation 2 for the full six year period. In the second, the water level in the reservoirs is reset to be equal to the observed reservoir level at the start of each year (Simulated Nudged). These simulations show that across the six years volume differences range from 1% to 30% of the actual

water storage at the start of each of the water years. Derived emulated water allocation decisions for GS show an underestimation of the allocation compared to the actual volume for most years, especially for the 2014water year. These differences occur because a constant factor is used (78%) to simulate the carry-over between water years. Results from the Simulated Nudged show how the daily water storage simulations and the water allocation for GS would behave using the actual water storage information (including the actual carryover volume), by nudging the simulated storage levels to the

observed at the start of each water year (Figure 5). The daily water storage simulations and water allocation to GS are now closer to the actual values, especially for the 2014water year. However, for both simulations, the emulated decisions show a similar trend when compared to the actual decisions. Upward revisions as well as where the water allocation remains constant (no revisions) occur at the corresponding announcement dates, and although results are slightly biased in volume, the emulation of allocated water decisions does follow the pattern of the actual decisions.

We apply the water allocation framework to simulate the water storage continuously over the 1982-2009 water years, assuming a constant allocation -use reduction factor of 0.78 for all water years. The framework is applied to compare water allocation decisions using different estimates of available water for allocation informed by seasonal forecasts of reservoir inflow, including the seasonal forecast datasets (FoGSS).

## 3.2 Performance of seasonal predictions of available water

Prior to applying the FoGSS forecasts to emulating the water allocation decisions, the reliability of the seasonal predictions of available water is evaluated. In Figure 6 and Figure 7 rank histograms are shown for both the POAMA and ESP+ datasets for the forecast ensemble of 1000 members (FoGSS) of the expected inflows into the Burrinjuck reservoir. The expected inflow





shown is the total inflow from the forecast month through to the end of the water year, which is the information on the available water that is required to inform the water allocation decision. Rank histograms for expected flows for the Goobarragandra River at Lacmalac show a similar pattern and are available in the supplementary material (Figure S12 and Figure S13). The rank histogram shows the frequency of the rank of the observed in the ensemble, with a well calibrated ensemble exhibiting a uniform distribution (Wilks, 2011). For easier interpretation, the ensemble is pooled into five classes. The light grey bar shows the frequency of the observed rank being higher (or lower) than the highest (lowest) forecast value in the ensemble.

The rank histograms of the expected inflows to end of season for forecasts made in the months from July until January show that the ensemble is under-dispersed as the distribution is increasingly U-shaped, with the POAMA dataset exhibiting better performance than the ESP+ dataset. The first three of these months (July to September) are the wetter season. As this recedes, the under-dispersion increases until December, after which the performance again improves, with the expected inflows in February showing a near uniform distribution, though reliability of the forecast again decreases as the accumulation period becomes shorter.

We additionally measure the accuracy of water year forecasts at each issue time with the continuous ranked probability skill score (CRPSS, Hersbach, 2000), with a climatology reference forecast generated by drawing random samples from a log-sinh transformed normal distribution (Wang et al., 2012) fitted to observations using the Bayesian Joint Probability model (Wang and Robertson, 2011). Positive CRPSS values indicate that FoGSS is more accurate, on average, than the climatology forecasts. Results show that skill is quite consistent for the forecasts flows of the Goobarragandra River, while the inflows of Burrinjuck Reservoir are only skilful for forecasts issued for July-September. In general, POAMA does have slightly better skill for forecasts issued earlier in the water year (See Supplementary material, Figure S11).

In summary, the reliability of the forecast ensembles is better between February and June compared to July and January, while forecast skill is better for the beginning of the water year (considering that the inflows to Burrinjuck are larger than the Goobarragandra River flows). In our study we are primarily interested in the predictions of the expected inflows from July to February to support water allocation decisions for the cropping season, which are made from November to February. For this period, the forecast ensemble is shown to be somewhat overconfident, especially for forecasts issued for December. In addition, the ESP forecasts are slightly negatively biased for Aug-Oct (i.e., they tend to underestimate inflow). How this affects the water allocation decisions, and if using the forecast ensemble leads to better estimates of available water compared to the currently used conservative estimate based on climatology, is evaluated using both the ESP+ and the POAMA forecasts to inform the water allocation decisions.

## 3.3 Water allocation using the seasonal forecast datasets

Water allocation decisions for General Security (GS) were emulated for the 1982-2009 period using four datasets of expected inflows to the reservoir to determine water availability: (i) observed inflow (considered as perfect information); (ii) the conservative inflow (or reference information as currently used by the decision maker); (iii) the FoGSS seasonal forecast based





on POAMA, and (iv) the FoGSS seasonal forecast based on ESP+. Water allocations using the perfect information and the reference information provide the benchmark against which the decisions informed by the ensemble forecasts are compared. For each of the two ensemble forecasts, two setups were tested. In the first, the inflow prediction at the beginning of the water year is obtained from the ensemble forecast made on July $1^{st}$, and this is then maintained for the next 12 months (non-updating

FoGSS set up). This means that the forecast of the available water that is established on July $1^{st}$ is not updated by newer forecasts as the water year progresses. This was done to mimic the current procedure used by the basin authority when using the conservative inflow estimate based on climatology, where the expected inflow is established at the beginning of the water year and then maintained to the end of the water year. In the second set-up that was tested, the full potential of the ensemble forecast is explored. The water availability estimate to the end of season is now updated each month using the FoGSS

forecasted inflows determined from the seasonal forecast made at the start of that month. In determining the expected water availability from the FoGSS forecasted inflows for both set-ups, different non-exceedance percentiles of the forecast ensemble are selected to provide the expected availability of water for allocation, starting with the $1^{st}$ non-exceedance percentile (commensurate with a very conservative estimate of water availability), and increasing this to the $50^{th}$ non-exceedance percentile (commensurate with the ensemble median).

### 3.3.1 Using one prediction at the beginning of the water year.

In this first setup, water allocation decisions were emulated using the inflow prediction obtained from the seasonal forecast at the beginning of the water year, and then not updated for the next 12 months (non-updating FoGSS set up). Figure 8 shows the water allocation decisions to General Security (GS) for a selected wet year (1998) and a selected dry year (2006) using the $1^{st}$, $5^{th}$, $10^{th}$, $25^{th}$ and $50^{th}$ non-exceedance percentile of the ESP+ and POAMA ensemble forecast datasets. The blue line shows

the allocation decisions made using the reference conservative inflow estimate, while the red line shows the allocation established using perfect information.

For the dry year of 2006, the results show that the allocation decisions using the forecast ensemble are similar to the decisions obtained with the conservative inflow for the $1^{st}$ percentile. This makes sense as the water allocation using the lowest percentiles is the water allocation that matches best with the water allocation based on the conservative inflow. It is interesting

to note that the water allocation based on perfect information is even lower than when using either the conservative inflow or the $1^{st}$ percentile. This is due to 2006 being the driest year on record (Dreverman, 2013), with observed inflows below the $1^{st}$ percentile. For the water allocations obtained with the $1^{st}$ percentile, as well as with the conservative inflow there are no downwards revisions during the water year. However, for increasing percentiles, the number of downward revisions increases as the initial estimate of available water at the start of the year becomes increasingly over confident. For this dry year, the

POAMA and ESP forecasts exhibit broadly similar behaviour.

In the wet year (1998), the initial allocation based on the $1^{st}$ percentile is higher than that using the conservative estimate, particularly for the POAMA dataset. For increasing percentiles, the water allocation decision approach those using perfect





information. For 1998, the observed water availability was well above the requirement to fulfil 100% of the allocation to General Security. No downward revisions are found, even for the higher percentiles.

These results provide an initial comparison between using the conservative inflow and forecasted inflow for determining the available water for allocation. However, for the full potential of the seasonal forecast to be evaluated, inflow predictions are
updated monthly during the water year as new FoGSS forecasts become available.

### 3.3.2 Updating the inflow prediction every month

In Figure 9 and Figure 10, the water allocation decisions are shown using the 1st, 5th, 10th, 25th and 50th non-exceedance percentiles of the ESP+ and POAMA forecast datasets. This uses the second setup, where the inflow predictions are updated every month with the most recent forecast information. Results for the dry years are shown in Figure 9, with those for wet
years shown in Figure 10. Of the 28 years evaluated, 13 are considered as dry (1982, 1994, 1997 and 1999-2009, the latter period constituting the millennium drought) and 15 as wet (1983-1996 and 1998). Wet years are taken to be those years when the final water allocation to the General Security attains 100% of the concession by the end of the season. For the dry years we show results for three years (1982, 2003, and 2006), with the results for the remaining years provided in the supplementary material (Figure S1 to Figure S9). The selected three years have different levels of water allocation based on perfect allocation
(70%, 55%, and 10% of the full concession), reflecting increasingly severe drought conditions. The reference water allocation based on the conservative climatological estimate is again shown in red. For the wet years we show results for three selected years (1988, 1995, and 1998), again for increasingly dry conditions. Results for the remaining years are again provided in the supplementary material.

For both dry and wet years, results found with POAMA and ESP+ differ only slightly in magnitude, and follow a similar trend
during the water year, though the median POAMA forecast predicting full allocation months earlier than ESP+. For most years, results using the forecast ensembles show that the derived water allocation decisions tend towards those established with the perfect water allocation. For all wet years and many of the dry years, the water allocations using the forecast ensemble are generally closer to those made using the perfect water allocation compared to those made with the reference water allocation. In the wet years results closest to the perfect forecast are obtained with the higher forecast percentile (less conservative
estimate), while for the dry years the lower percentiles provide the best results.

For some of the dry years, particularly for 1982 and 2006 (Figure 9), the water allocation using the conservative inflow initially overestimates the "real" water allocation based on perfect information, and is then revised downwards as the season progresses. In 2006, the water allocation based on the conservative inflow is 10% higher than the perfect water allocation during the entire year. Observed inflows for the 2006 water year were the lowest on record (Dreverman, 2013), and thus lower than the
conservative estimate. Similar behaviour is found in 2007 and 2008, though the initial storages at the start of these dry years was already so low that allocations never exceeded 0%. The overestimation at the start of the season may well be attributed to a wet bias in the forecast for these dry years, with the more conservative forecasts (1% - 10%) thus providing the best estimate of actual inflows. In the less extreme dry years, the water allocation using the conservative inflow is lower than the perfect




water allocation at the beginning of the water year and progressively increases (e.g.1982 and 2003). Using the forecast ensemble shows better water allocation results compared to when using the conservative inflow. For many of the dry years (see Figure 9 and supplementary material), downward revisions of the water allocation do, however, occur during the water year for all percentiles of the forecast ensemble. As expected, selecting a higher forecast percentile to establish the expected

inflow leads to a higher water allocation at the beginning of the water year, and consequently to larger downward revisions, though the difference in water allocations between forecast percentiles converges as the year progresses. In some years (e.g. 2003 and 2006) the magnitude of water allocation using the forecast ensemble is similar to that of the perfect water allocation, though this depends on which date the water allocation is estimated and which forecast percentile is used. For example, in the dry year of 2006 the initial water allocation (July 1st) using the forecast ensemble is overestimated (at 15% for the 1st percentile

of ESP+), but as of September 14th the predicted water allocation tends towards the perfect water allocation at 9%. A similar trend happens for the dry year of 2003, but in this case the initial water allocation using the forecast ensemble is low (at 40% for the 1st percentile of ESP+) and then trends towards the perfect allocation at 54%. For the dry year of 1982 the water allocation using the forecast ensemble does not tend towards the perfect water allocation, at least not for the 1st percentile. Using the 50th percentile the water allocation initially tends towards the perfect water allocation on September 14th, but the

water allocation is subsequently underestimated, leading to continuous downward revisions. For those wet years (Figure 10) where the initial water estimate is below 100% (e.g. 1995 & 1998), there are primarily upward revisions of the allocation (with only sporadic downward revisions) for all percentiles of the forecast ensemble. Water allocation results using the forecast ensemble are generally equal to the perfect water allocation after the September 14th decision date for all non-exceedance percentiles and remain so until the end of the cropping season. For several wet years, the water allocation decision based on

the conservative inflow are lower than for the perfect water allocation at the beginning of the water year and then progressively increase (e.g.1982 and 2003). These may be up to 55% lower than the perfect water allocation of 100% (e.g. 1998). For all years where this occurs, using the forecast ensemble shows better water allocation, even for the lowest forecast percentiles, which are closest to the conservative forecast. For many of the wet years, the initial allocation using both the perfect information, as well as the conservative inflow is already at 100%, as are allocations based on all forecast percentiles.

Figure 11 shows the inconsistency indices for all 28 years tested, using the configuration with the POAMA forecast, where the forecast data is updated each month as a new forecast becomes available. In the supplementary material in Figure S10 the inconsistency indices using the ESP+ forecast is shown. The years are ranked in order of the observed inflow volume, with 2008 having the lowest inflow volume and 1983 the highest. Years marked in red are considered as dry while those in blue are considered wet. The annual inconsistency of the allocation due to upward revisions (positive inconsistency) for the different

forecast percentiles, as well as for the conservative flow is shown in figure (a), with the downward revisions (negative inconsistency) shown in figure (b). The figure shows that using a higher non-exceedance percentile to inform the water allocation decision leads to less upward revisions (lower positive inconsistency) for all years, and in all cases provides an improvement over the allocation based on the conservative inflow estimate. This is seen primarily for the wetter dry years and the dryer wet years. For wetter wet years there are no upward revisions as the allocation already starts at 100% of the





entitlement, while for the more extreme dry years there are also no upward revisions of the water allocation due to the sustained lack of water. The reduction in upward revisions does, however, come at the cost of more frequent downward revisions (higher negative inconsistency) for the drier years, particularly for the more extreme drier years and for the higher percentiles. Although the annual inconsistency provides information on revisions of the water allocation during the water year, it does not

allow for easy comparison against the perfect water allocation. Figure 12 shows the Root Mean Squared Difference (RMSD) calculated over the 28 years using the difference between the water allocation established using the ensemble and that of the perfect forecast for each decision date, both for the dry and wet years, as well as the forecasts using the ESP+ and the POAMA datasets. The RMSD calculated using the conservative inflow estimates are also shown. An RMSD value of zero would imply the perfect allocation. For dry years, the RMSD using the monthly updated forecast ensemble shows lower differences than

the reference conservative water allocation for all non-exceedance percentiles until the end of October for both ESP+ and POAMA, with the latter marginally outperforming the former. This is also the case for the wet years, though using lower percentiles for the forecast leads to higher differences in allocation than when considering the conservative inflow forecast. It is important to note that for allocation decisions informed by either of the forecast datasets, there is a major error reduction between August 30th and September 14th. After September 14th the difference remains more or less equal until the end of the

cropping season. This is significant, as it means that farmers will have clearer information on their allocation several weeks earlier than is the case when using the conservative inflow forecast.

## 4 Discussion

The water allocation framework developed in this study was applied to assess the benefit of using a seasonal forecast ensemble (FoGSS) in estimating the available water for allocation. The potential for farmers to benefit from this seasonal forecast is

discussed from three perspectives; i) how well climate can be predicted, ii) to what degree this information helps in the actual decision process and iii) to what extent it contributes in reducing the negative impacts (Hansen, 2002).

### 4.1 How well can the inflows be predicted?

FoGSS is an experimental seasonal ensemble streamflow forecast for a 12 month horizon developed for the Australian continent (Bennett et al., 2016, 2017) . FoGSS post-processes climate forecasts, either derived from ESP+ or POAMA to force

a monthly hydrological model. ESP+ is an ensemble of seasonal precipitation forecasts based on climatology, while POAMA is an ensemble coupled ocean-atmosphere general circulation model (CGCM).
A long time series (28 years) of climate predictions was used assuming a full representation of climate variability. The time period includes extremely dry years between 2001 and 2009 referred to as the Millennium Drought (van Dijk et al., 2013) and very wet years, such as 1988 and 1989 (BoM, 2019).

The derived inflow predictions were transformed into accumulated inflow volume from each prediction month until the end of the water year (June). This set up was used to mimic the current water allocation process in which the basin authority uses



a conservative inflow prediction at the beginning of the water year for the next 12 months. In our approach a prediction of the inflow to the end of the season is updated each month using the forecast ensemble as a new forecast becomes available. Our approach to verifying the FoGSS seasonal forecasts is a novel extension of the traditional forecast skill assessment in that inflow predictions evaluated for the next *n* months until the end of the water year, representing the decision variable used in

the water allocation framework. In addition, key decision dates for water allocation are evaluated to provide insight into the reliability of those forecast that have impact on farmers planning and operational decisions, especially for months before and during the cropping season.

As the seasonal forecast is updated each month, with a progressively shorter lead time to the end of season, we would expect progressively more accurate forecasts of the water year. This is, however, only partly reflected in the inflow forecast skill

results. Inflow forecasts to Goobarragandra are skilful year-round, with a slight dip for forecasts issued around December, and a rise at the end of the water year (May-June) as the accumulation period shortens. FoGSS Forecasts for Burrinjuck tend to be less accurate, and skill varies strongly through the year: skill is high for forecasts issued beginning of July and August, but decreases every month from July until December, after which it again improves. The accuracy of Burrinjuck forecasts is thus clearly a function of season. For predicted forecasts issued in July-September (towards the end of the wetter winter season)

the accuracy of accumulated volume forecasts is dominated by initial hydrological conditions and information from recent (high flow) months. Flows are much lower by December, meaning that flow volumes in accumulated inflow forecasts are dominated by higher flows late in the water year (April-June). Predicting the rise of the annual hydrograph accurate relies on rainfall forecasts, which are usually not skilful at longer lead times. Thus water year forecasts issued for Burrinjuck in the driest months (December-February) tend to have the poorest skill. Inflow predictions to Burrinjuck after February do

progressively improve due to the shorter lead time of the prediction, as the updated climate forecasts compensates the model uncertainty.

In both catchments, the use of calibrated POAMA climate forecasts to force FoGSS adds skill to inflow forecasts issued in July-September. Schepen et al. (2014) reported positive skill for calibrated POAMA forecasts from July-October, broadly coinciding with this period. While overall seasonal climate forecasts may not always add skill to inflow forecasts (Arnal et al.,

2018; Bennett et al., 2017), we show that for these catchments seasonal climate predictions offer a small but marked improvement in forecast accuracy. The use of calibration to ensure coherent climate predictions – i.e., as skill declines with lead time climate forecasts revert to climatology - ensures these gains in skill are not lost through poor climate forecasts at longer lead times.

## 4.2 To what degree does the seasonal forecast help in the decision process?

Currently, only upward water allocation revisions are made during the water year in river basins in Australia. This is because in the current allocation a conservative inflow for the next 12 months based on climatology is used to determine the water allocation. Essentially this results in an estimate of available water that is always on the safe side from the point of view of the allocation decision. We estimate that the conservative estimate used in the current policy equates to the inflow that is exceeded





97% of years, which would imply that in virtually all years the actual inflow is higher than the initial estimate. Depending on the initially available water in the storages, this results in upward revisions of water allocations as the season progresses for most years, with the exception of the most extreme dry years (such as 2006). To the farmer, advance knowledge of the ultimate allocation is beneficial as it allows for better planning. We show that using information from a seasonal forecast ensemble to

predict the inflows to the reservoirs and inform the water allocation decision can reduce the upward revisions. Although using the seasonal forecast ensemble does not reduce the frequency of upward revisions (or positive inconsistency) in all years, it does improve the accuracy in estimating water available for allocation. This implies there is benefit to using the seasonal forecast ensemble to inform the decision process. However, this comes at a cost. There is a trade-off between obtaining better predictions of water available for allocation than the conservative low estimate as it comes at the cost of more downward

decisions during the water year. This is evident in dry years, particularly where the positive bias in the forecast at the start of the season results in downward revisions of the water allocation as the year progresses. In wet years upward revisions are reduced (compared to the reference water allocation) and only very minor downward revisions occur. Despite this, for both dry and wet years the accuracy of the available water for allocation using the forecast ensemble improves during the water year. This is most evident on the decision date around September 14th, where the accuracy of the water allocation decision

informed by the forecast improves significantly and maintains this accuracy until the end of the water year. This is evident from the results of the root mean square difference (RMSD), which shows the magnitude of the difference in the allocation decision through the season when compared to that made with the perfect knowledge of the amount of water available. The root mean square difference with the forecast ensemble attains a value more or less equal to the root mean square difference obtained with the conservative inflow, but two months before the original decision date (moving forward from about November

13th to September 14th). This means that for the same accuracy of predicted water allocation, decision makers can rely on the forecast ensemble information two months in earlier when compared to the conservative inflow information. This may be of significant benefit to the farmers as they can then better plan their irrigation season based on the amount of water they would expect to be allocated.

Whether the basin authority in charge of water allocation announcements would choose to adopt a seasonal forecast system

such as FoGSS and change the water allocation policy will depend very much on how acceptable downward revisions of allocated water are. The current policy has been designed to avoid such downward revisions unless exceptional circumstances dictate. Using the full potential of the seasonal forecast (including a monthly update of the expected inflow as new forecasts become available) provides more accuracy in water allocation estimates at the cost of downward revisions. Following maximum utility theory, the acceptability of downward revisions will depend on the impact these have, compared to the benefit

of the improved and earlier information on the water allocation. The ratio of downward and upward revisions is also influenced by the selection of the non-exceedance percentile. For dry years, selecting a lower, more conservative, percentage would appear to be the best strategy, while for the wet years a higher percentage should be selected. A non-exceedance percentile of 10% appears to provide the best performance on average for both dry and wet years evaluated in this study, but a more dynamic approach could also be taken depending on the forecast as well as the available storage at the beginning of the water year. It



would appear that the downward revisions (that occur mainly in the dry years) are primarily due to biases in the inflow forecast. Comparing the results of allocation decisions informed by the FoGSS forecasts based on the POAMA dataset to those based on the ESP dataset, the number of revisions (both upward and downward) in Figures 11 & S10 indicate that these are marginally less for the POAMA based dataset. This would suggest that further improving the seasonal forecast can contribute to reducing

undesirable downward revisions. Additional improvements to the inflow predictions through reducing the uncertainty in the hydrological model of the basin will also contribute to reducing the bias of the inflow predictions and improving allocation results.

## 4.3 To what extent does the seasonal forecast contribute to reducing negative impacts?

The impact in irrigated agriculture of uncertainty in the available water resources has been widely assessed in Australia

considering climate variability and climate change scenarios (Kirby et al., 2013, 2014a, 2014b, 2015). Adaptation measures, reallocation strategies and policy reform are currently in discussion to prevent future impacts due to extreme events (Bark et al., 2014; van Dijk et al., 2013). In this study we explore the possibility of using a seasonal forecast ensemble to secure the right amount of allocated water at the right time during the water year. In the Murrumbidgee basin, farmers decide on the area to be cropped for annual crops based on the water allocation announcement for General Security. The water year starts on July

1$^{st}$, but the summer cropping season starts on November 1$^{st}$ and ends March 1$^{st}$. In that sense, the period to decide on the area to be cropped is between July 1$^{st}$ and November 1$^{st}$, and the period for operational decisions (e.g. irrigation schedule, weed management) is from November 1$^{st}$ to March 1$^{st}$. While it would seem logical that farmers wait until the last allocation announcement before November 1$^{st}$ to decide on the area to be cropped, due to pre-cropping planning activities and investments (e.g. buying seeds, maintenance of irrigation assets, or investing in agricultural equipment and machinery) they prefer to take

decisions earlier, therefore relying on water allocation announcements made at an earlier date. Through using the seasonal forecast to inform water allocation decisions we show that famers could rely on the water allocation announcement made on September 14$^{th}$, some 1 ½ months earlier. This would allow them to plan their activities better, thus reducing potential negative impacts of having to take decisions on the area to be cropped before the actual water allocation on November 1$^{st}$ is established.

## 5 Conclusion

We apply a water allocation framework to assess the benefit of using a seasonal forecast ensemble to inform water allocation decisions. This water allocation framework uses an estimate of the available water for the irrigation season that is based on the balance of the demand to the available water in the reservoirs in a basin and the expected inflows to those reservoirs from the decision date until the end of the water year. The water allocation framework emulates current water allocation policy, following which the basin authorities make decisions on the allocation of water to meet claims as defined in water concessions.

Depending on availability, water may be allocated to fully meet these concessions or only to a set percentage. We apply the framework in the Murrumbidgee basin in Australia. In this basin, conservatively low estimates of the expected inflow based



on climatology are currently used at the beginning of the water year to estimate water available for allocation. As the water year progresses, water allocated to each concession may be revised if expected water availability changes. As the initial estimates are conservative, water allocations are mostly conservatively low, and consequently for the majority of years are revised upwards as the season progresses. Although upward revision of the allocation is beneficial to irrigators, advance and

consistent information on the water allocated is important to them to help better plan their irrigation season.

Instead of the currently used conservative low estimates for inflow predictions we propose using inflow predictions from an ensemble seasonal streamflow forecast to inform water allocation decisions. Inflow predictions are obtained from the "Forecast Guided Stochastic Scenarios" (FoGSS), an experimental 12-month ensemble streamflow forecasting system using either historical rainfall sequences (ESP+, an extension of the Ensemble Streamflow Prediction; ESP approach) or the POAMA M2.4

seasonal climate forecasting system as climate forcing. Of the two, predicting the inflows using the POAMA datasets were found to have better skill than using the ESP+ datasets, though both exhibit seasonal bias. Applying the water allocation framework to emulate decisions made for 28 years (from 1982 through 2009) shows that the seasonal forecast ensemble helps improve the decision process as the water expected to be available for the water year is better predicted when compared to using the reference conservative forecast. However, overconfidence in the seasonal forecast may lead to overconfidence in the

expected availability of water. This may result in downward revisions of water allocation as the season progresses due to too high an allocation decision earlier in the season. This is more evident for dry years than it is for wet years, with downward revisions occurring more frequently than is currently the case. In wet years the number of upward revisions are reduced (compared to the reference water allocation), with virtually no downward revisions. Using the FoGSS seasonal forecast that is currently available would imply a trade-off to be established between the obtaining of a better estimate of the available water

and the cost of an increased number of downward revisions during the water year. Comparison of the FoGSS forecast based on POAMA and that based on ESP+ shows the former to be marginally superior, suggesting that further improvement of the seasonal forecast would further help improve allocation decisions.

For both dry and wet years, the accuracy of the available water estimates using the forecast ensemble improves progressively during the water year, with a particular improvement some one and a half months before the start of the cropping season in

November. This additional time is important to irrigated farmers, as it allows them to better plan the cropping season (November to February). Using the forecast ensemble thus benefits water allocation decisions established by the basin authority, allowing the final allocation to meet concessions to be determined more accurately and earlier in the season, resulting in a reduction of agricultural losses as a results of climatic variability.

**Acknowledgments**

This work received funding from the European Union Seventh Framework Programme (FP7/2007-2013) under grant agreement no. 603608, Global Earth Observation for Integrated Water Resource Assessment (eartH2Observe). JB is supported by the WIRADA partnership between CSIRO Land & Water and the Bureau of Meteorology, and ARC linkage project





LP170100922. We would like to thank CSIRO Land and Water in Canberra, Australia for sharing knowledge, information and data. Special thanks to Jorge Peña, Juan Pablo Guerschman and Marc Kirby.

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


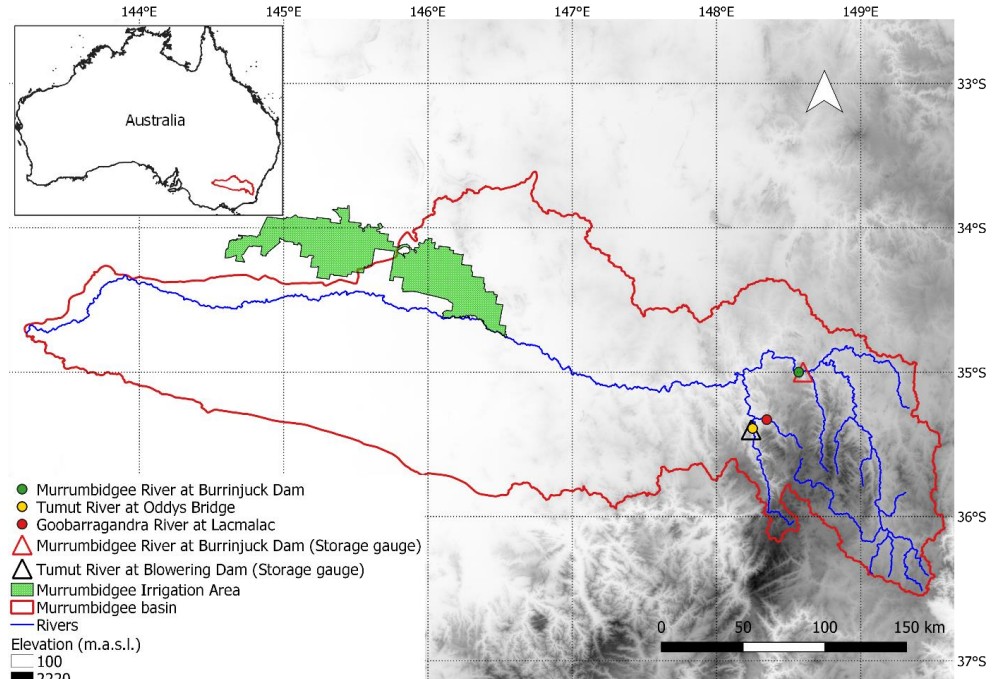

**Figure 1. Map of the Murrumbidgee River basin in Australia. Location of the Murrumbidgee Irrigation Area, discharge stations (Murrumbidgee River at Burrinjuck Dam, Tumut River at Oddys Bridge, and Goobarragandra River at Lacmalac) and storage gauges in the Burrinjuck and Blowering reservoirs. Elevation map is obtained from http://srtm.csi.cgiar.org.**

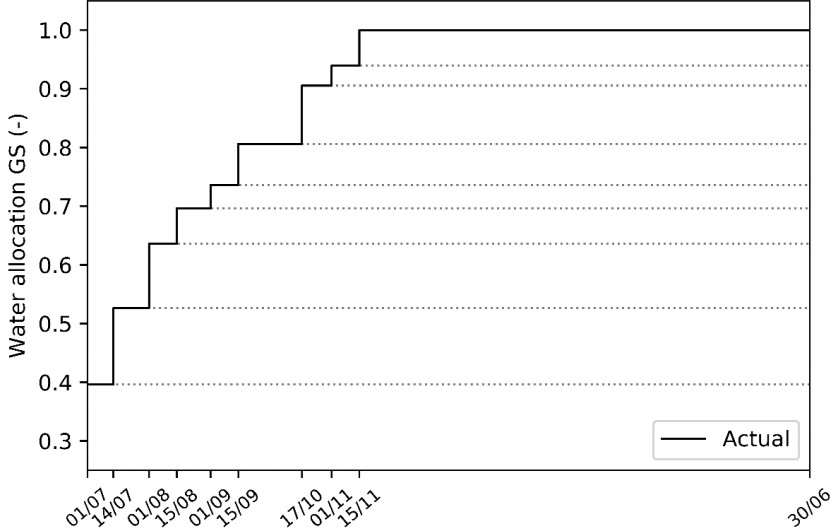

**Figure 2. Actual water allocation for General Security, GS in the Murrumbidgee basin for the water year 2016. Dotted lines in the water allocation curve are used to show that the water allocation is an annual water volume, the estimate of which changes during the water year.**





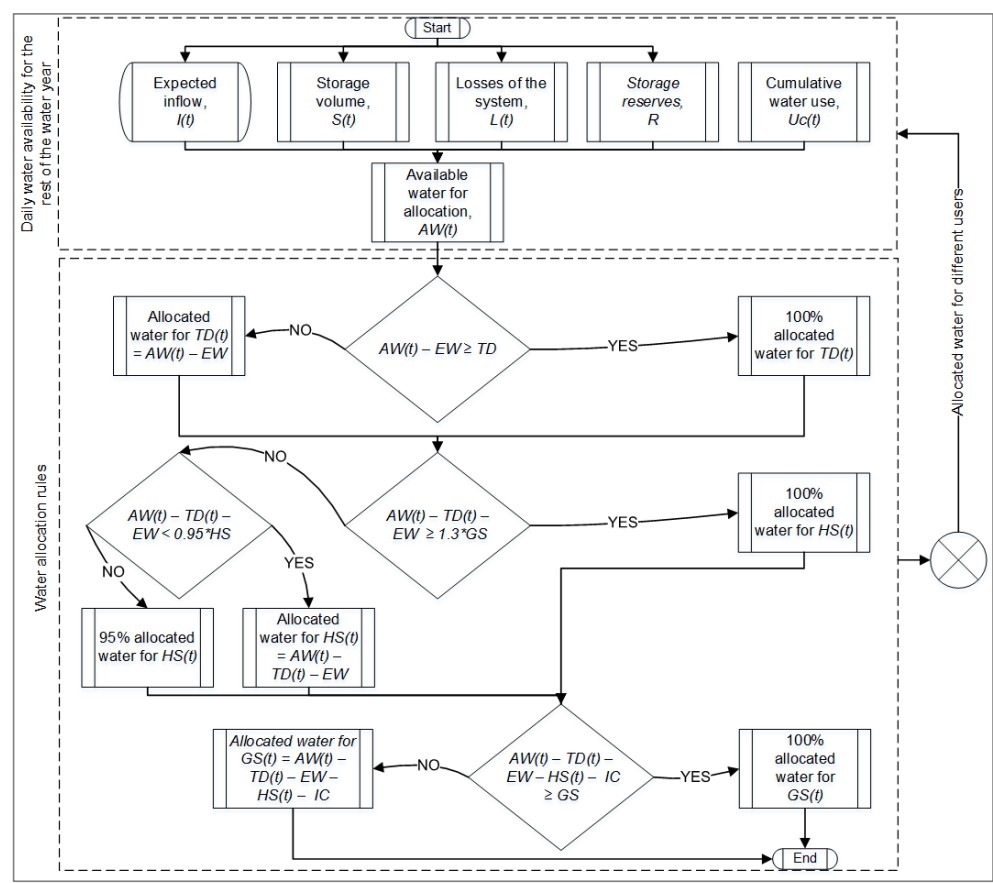

**Figure 3. Water allocation framework used to simulate the daily water availability for the rest of the water year and emulate the allocated water for different users based on allocation rules in the Murrumbidgee basin. The input variable is the expected inflow into the reservoirs.**



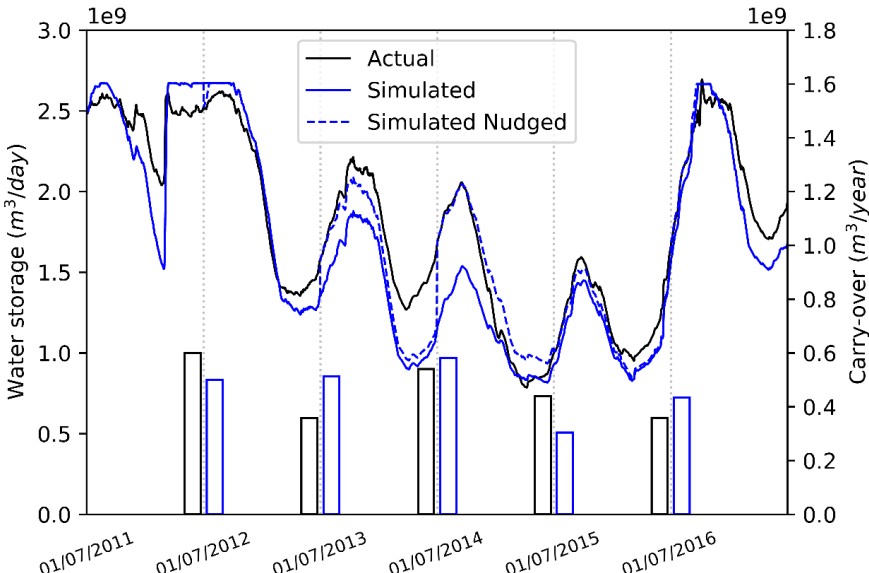

**Figure 4. Actual and simulated reservoir storage and carry-over volumes in Blowering and Burrinjuck reservoirs (2011-2016).**

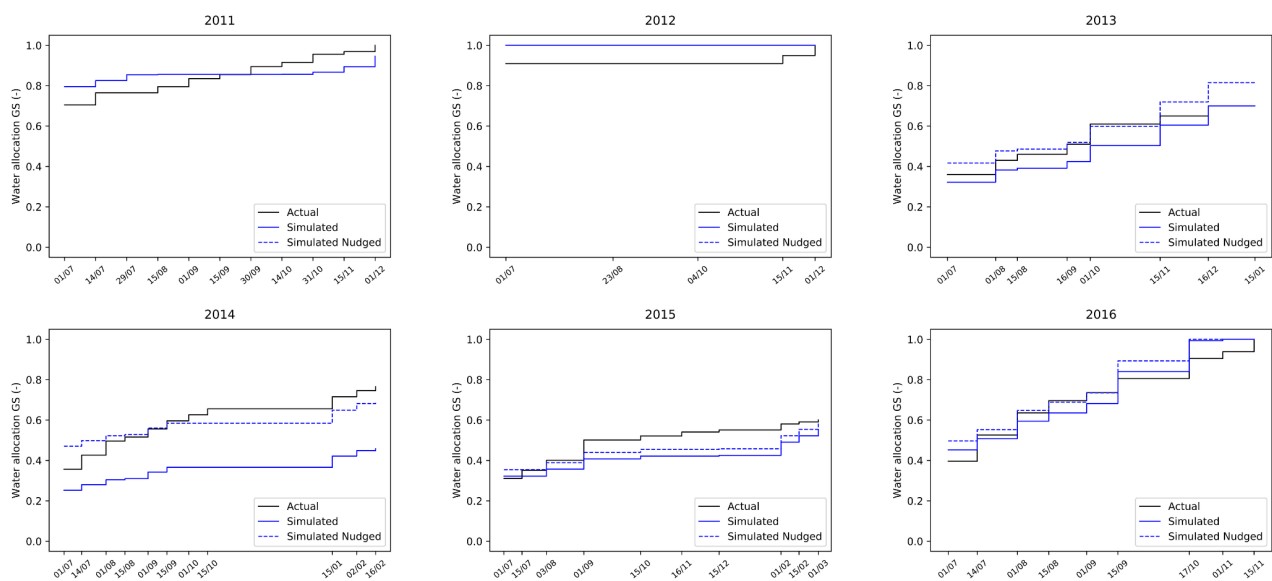

**Figure 5. Actual and emulated water allocation GS for each year (2011-2016) using conservative inflow. Simulated Nudged is the simulated water allocation GS, but using the actual storage on July 1st.**





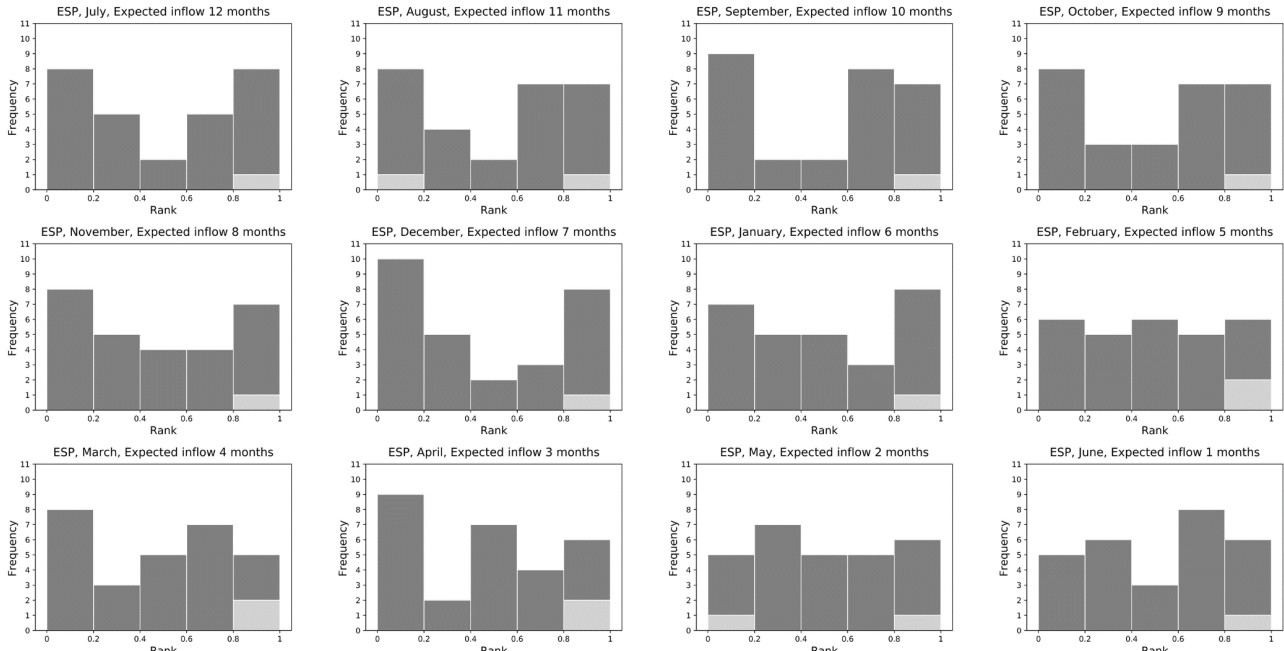

**Figure 6. Rank histogram using ESP datasets from FoGSS (1982-2009) for expected inflow in the next n months (Starting July) in the Burrinjuck reservoir.**





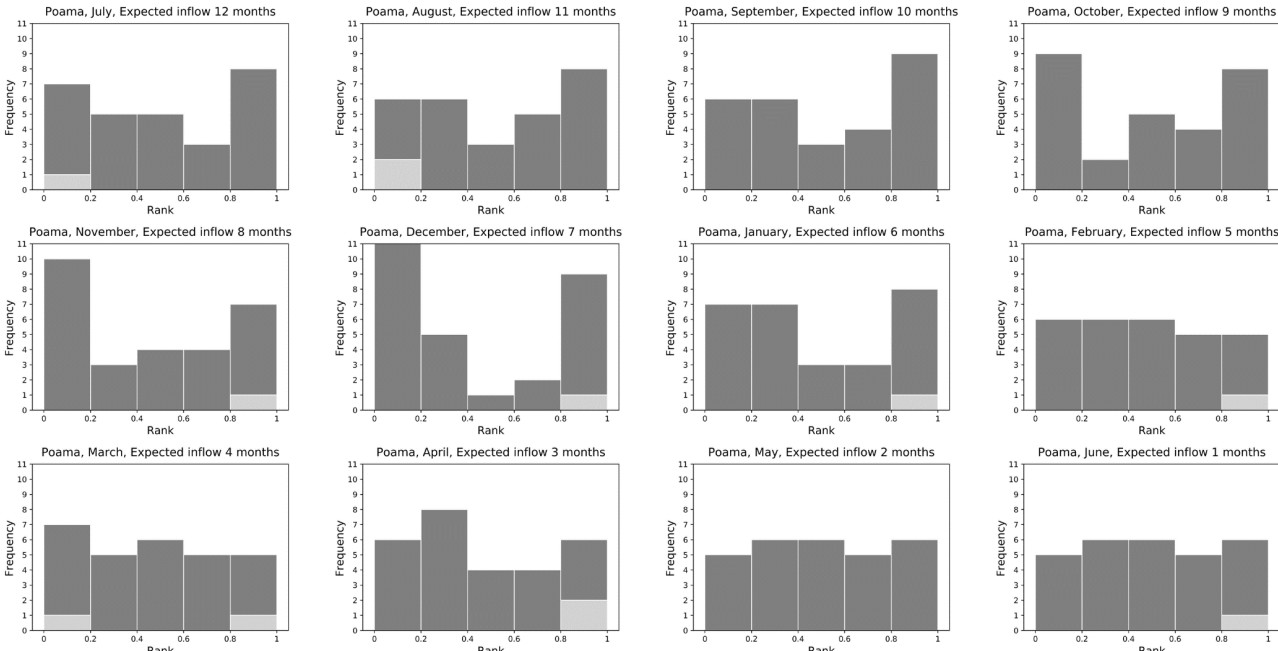

**Figure 7. Rank histogram using Poama datasets from FoGSS (1982-2009) for expected inflow in the next n months (Starting July) in the Burrinjuck reservoir.**



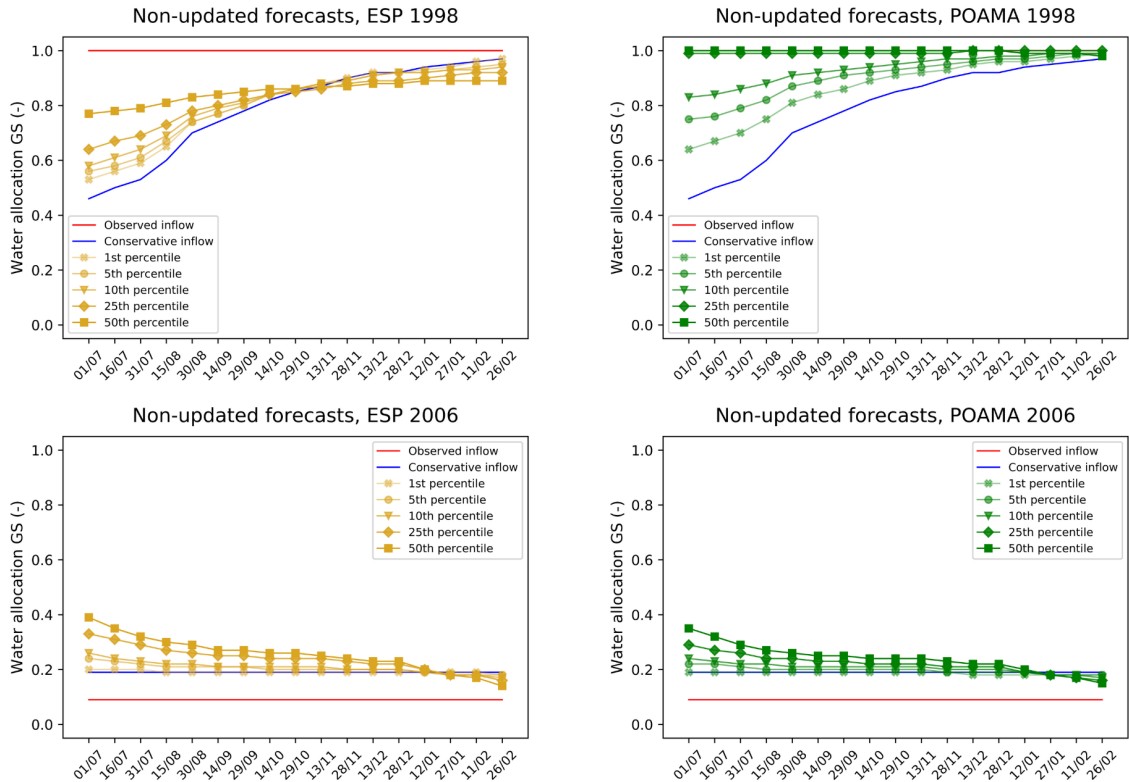

**Figure 8. Water allocation GS for one wet year (1998) and one dry year (2006) using one inflow prediction at the beginning of the water year for the next 12months (Non-updated forecasts)**



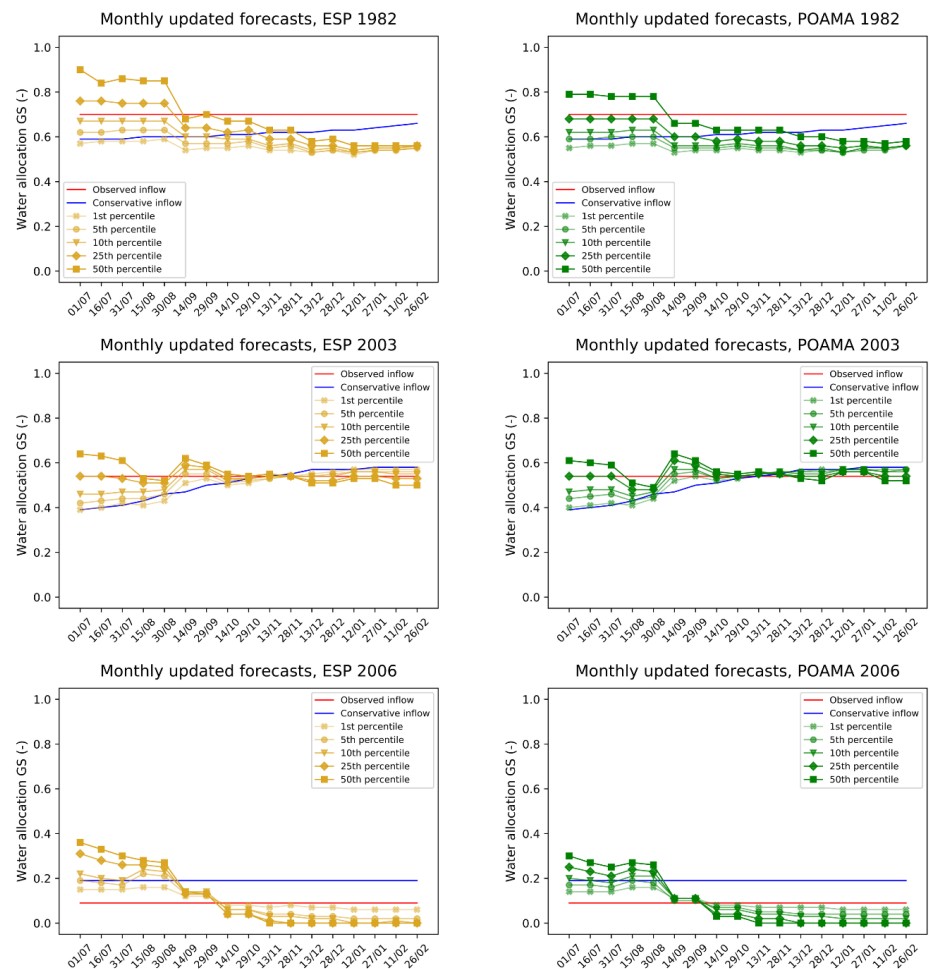

**Figure 9. Water allocation GS for selected dry years (1982, 2003 and 2006) using new inflow predictions every month (Monthly updated forecasts).**

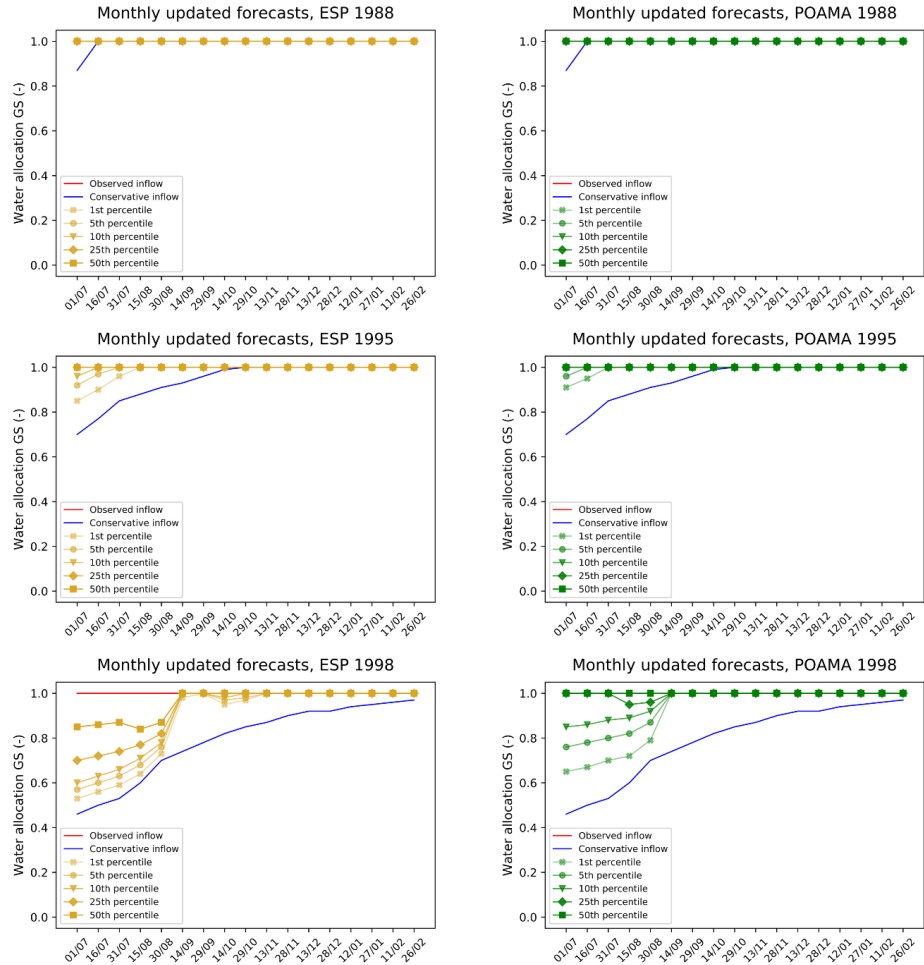

**Figure 10.** Water allocation GS for selected wet years (1988, 1995 and 1998) using new inflow predictions every month (Monthly updated forecasts).



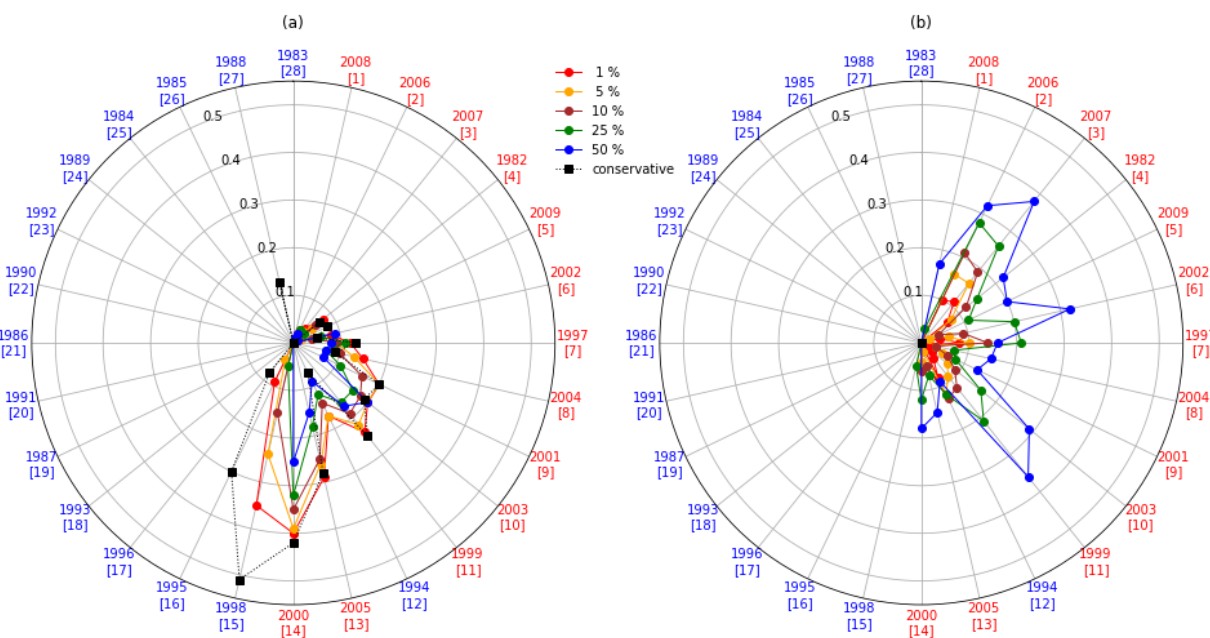

**Figure 11. Inconsistency for all 28 years using FoGSS/POAMA forecast updated every month to inform the allocation decision. (a) shows the annual inconsistency due to upward revisions (positive inconsistency) and (b) the annual inconsistency due to downward revisions (negative inconsistency).**



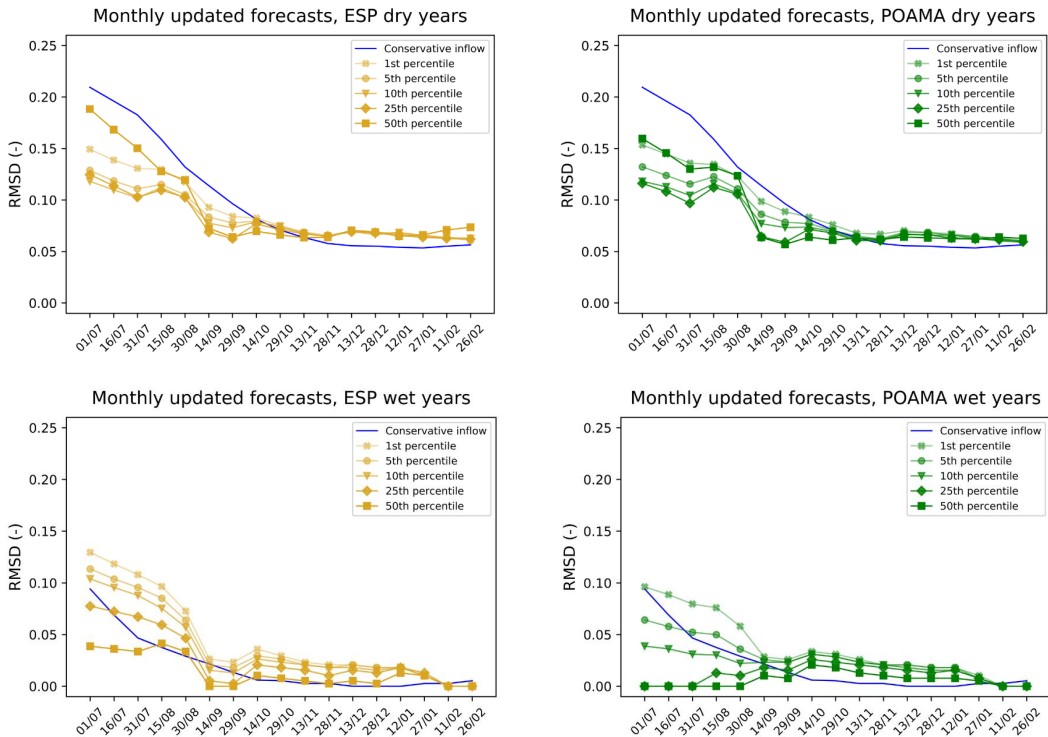

**Figure 12. RMSD for all dry years (upper two figures) and wet years (lower two figures) using new inflow predictions every month from ESP (left two figures) and POAMA (right two figures)**