# Peer review of "The benefit of using an ensemble of seasonal streamflow forecasts in water allocation decisions"

_Hydrology and Earth System Sciences, 2020_

## Referee Comment (RC1) · Anonymous Referee #1 · 19 Mar 2020

Streamflow forecasts can substantially contribute to the efficiency of water resources system. Yet, the efficiency improvements are influenced by multiple factors, such as the quality of forecasts, the characteristics of the system, and the optimization/simulation models for system operation. This paper has conducted an investigation of the use of ensemble seasonal streamflow forecasts in water allocation decision-making. The forecasts are generated by the FoGSS model; the water allocation is through a simulation model (Figure 3 on Page 23); and the case study is for the Murrumbidgee basin in Australia. In general, the paper is interesting with the results and methods clearly presented.

There are a few comments for further improvements of the paper.

First of all, is it possible to conduct the system operation using perfect forecasts? If so, the performance under perfect forecasts would serve a benchmark, i.e., upper bound, in the analysis. Specifically, it would illustrate the maximum benefit from the use of forecasts. In the analysis, the paper has analyzed the performance under climatology. Conceptually, it shows the lower bound of the system performance when no forecasts are available. The gap between the lower and upper bounds would highlight the potential benefit due to the use of streamflow forecasts.

Secondly, the system is operated by a simulation model. According to Figure 3 on Page 23, the model involves a number of thresholds that are quite empirical. How are the thresholds determined for the case study? How is the sensitivity of the system performance to the thresholds? Are the values of the thresholds optimal (or sub-optimal)? Details on the setting of the simulation model, in particular the empirical thresholds, ought to be provided.

Thirdly, according to Figure 4, the actual water storage tends to be higher than the simulated storage. On Page 9, the difference is related to that "a constant factor is used (78%) to simulate the carry-over between water years". This result highlights that the empirical thresholds can considerably influence the water allocation decisions and the performance of the water resources system. Is it possible to test the optimality (or rationale) of this critical threshold (and also other thresholds) of the simulation model?

Fourthly, stochastic optimization models are usually set up for reservoir operations using streamflow forecasts. For some examples, please refer to Labadie (2004, https://doi.org/10.1061/(ASCE)0733-9496(2004)130:2(93)), Celeste and Billib (2009, https://doi.org/10.1016/j.advwatres.2009.06.008), Zhao et al. (2012, https://doi.org/10.1029/2011WR010623), Turner et al. (2017, https://doi.org/10.5194/hess-21-4841-2017), and Anghileri et al. (2019, https://doi.org/10.1029/2019WR025280). One remarkable advantage of stochas-

tic optimization models is the explicit handling of forecast uncertainty. Also, the system performance can be optimized, instead of being simulated. For the case study, is it possible to set up an optimization model?

---

## Referee Comment (RC2) · Anonymous Referee #2 · 2 Apr 2020

This manuscript presents the benefit of using an ensemble of seasonal streamflow forecasts in water allocation decisions with an emphasis on those decisions in dry seasons and dry years. This is very important for farmers to choose which crop to plant and to decide on the area to be cropped. And also, the manuscript described the development of new approaches for the reservoir inflow estimates to replace the fixed inflow with the forecasted inflows, decision model to emulate the feedback loop between simulated reservoir storage and water allocation to irrigated crops, inflow forecasts, etc. The authors have briefly evaluated the approaches and identified effectively, and find that there is a quite much higher inconsistency and lower accuracy in estimating water available for allocation during dry seasons and dry years. This is a good and new

insight of present manuscript to enhance our understanding of the water allocation for the farmers. The subject is relevant to the journal, the manuscript is well written and structured. However, at present, the focus of manuscript is not particularly strong and it seems that the authors are not entirely sure about the key message they wish to convey. There are some aspects are suspected as follows: Firstly, the equations (on pages 6-8) to determine the available water for allocation needs more variables related to complicated relationships among the water demands and feedback loop among the reservoirs. Secondly, it is necessary to discuss the nonlinear processes of higher water demands and tradeoffs among the water users and reservoirs behind dams in the study area in dry seasons and dry years. These processes are suggested to presented more in detail in the context "4.2 To what degree does the seasonal forecast help in the decision process?". Thirdly, the better quality of figures in the text and supplementary materials are suggested to provide.

The manuscript is recommended to be accepted after minor revision.

---

## Author Comment (AC1) · 29 Apr 2020

We thank the reviewer for taking the time to review the manuscript and for the helpful comments and suggestions. Here we provide answers to the specific comments as well as indicating how we propose to improve the manuscript to address the issues raised by the reviewer.

General comments Streamflow forecasts can substantially contribute to the efficiency of water resources system. Yet, the efficiency improvements are influenced by multiple factors, such as the quality of forecasts, the characteristics of the system, and the optimization/simulation models for system operation. This paper has conducted an

investigation of the use of ensemble seasonal streamflow forecasts in water allocation decision-making. The forecasts are generated by the FoGSS model; the water allocation is through a simulation model (Figure 3 on Page 23); and the case study is for the Murrumbidgee basinin Australia. In general, the paper is interesting with the results and methods clearly presented. There are a few comments for further improvements of the paper.

1. First of all, is it possible to conduct the system operation using perfect forecasts? If so, the performance under perfect forecasts would serve a benchmark, i.e., upper bound, in the analysis. Specifically, it would illustrate the maximum benefit from the use of forecasts. In the analysis, the paper has analyzed the performance under climatology. Conceptually, it shows the lower bound of the system performance when no forecasts are available. The gap between the lower and upper bounds would highlight the potential benefit due to the use of streamflow forecasts.

Reply: We agree that using the perfect forecast serves as an upper bound benchmark for assessing the performance of forecast information, with the performance under climatology soften used as a lower bound benchmark. We would like to underline that we have indeed used observed inflows into the reservoirs as perfect information forecasts. This is clearly outlined in the methodology (starting line 30 on page 10): "Water allocation decisions for General Security (GS) were emulated for the 1982-2009 period using four datasets of expected 30inflows to the reservoir to determine water availability:(i) observed inflow (considered as perfect information); (ii) the conservative inflow (or reference information as currently used by the decision maker); (iii) the FoGSS seasonal forecast based on POAMA, and (iv) the FoGSS seasonal forecast based on ESP+.

As suggested by the reviewer, we also use the perfect forecast information as a benchmark, e.g. in the calculation of Root Mean Square Difference (ðİŚĚðİŚĂðİŚĘðİŘů) statistic, which was used to evaluate the performance of the allocated water obtained with the expected inflows against the allocated water obtained with the observed inflows (perfect information) for dry and wet years. As a lower bound for evaluating the

performance of the decision model as informed by the seasonal forecast we use the conservative inflow prediction. This is based on climatology and is in our opinion a useful benchmark as it is used in the current policy. For the evaluation of the seasonal inflow predictions provided by FoGSS we do apply a lower bound reference based on climatology, as explained on page 10, line 8.

2. Secondly, the system is operated by a simulation model. According to Figure 3 on Page 23, the model involves a number of thresholds that are quite empirical. How are the thresholds determined for the case study? How is the sensitivity of the system performance to the thresholds? Are the values of the thresholds optimal (or sub-optimal)? Details on the setting of the simulation model, in particular the empirical thresholds, ought to be provided.

Reply: In the water allocation model that we develop we have used parameters and thresholds based on established water allocation policy and that is used in the basin to establish water allocation decisions (e.g. volume of water allocated to the environment or estimated to be lost in conveyance). We agree that the sensitivity of these thresholds can be evaluated, as these will influence the allocation to general security users, which is the focus of this study. However, our objective is to evaluate the benefit of using the ensemble of seasonal streamflow forecasts to inform decision made within the context of the established water policy and regulations. Changing these parameters and thresholds would constitute a change to the policy. Further research could explore how changing water policy and regulations through changing thresholds influences the allocation decisions and what may be optimal values. As our intent is to explore the benefit of seasonal forecasts in informing decisions made in the context of the current policy, we apply the thresholds as defined in the existing policy. To clarify the origin of the thresholds, we propose to include the comment: "Note that decision thresholds are derived from the existing water allocation policy and regulations".

3. Thirdly, according to Figure 4, the actual water storage tends to be higher than the simulated storage. On Page 9, the difference is related to that "a constant factor is

used (78%) to simulate the carry-over between water years". This result highlights that the empirical thresholds can considerably influence the water allocation decisions and the performance of the water resources system. Is it possible to test the optimality (or rationale) of this critical threshold (and also other thresholds) of the simulation model?

Reply: We agree that this factor has an important influence on water allocation decisions as it determines the fraction of the amount of water allocated to farmers that they actually use. The constant factor value of 78% was obtained based on a performance evaluation between simulated carryover volumes and observed carryover volumes for the 2011-2016 period (this period was selected as prior to that a different water allocation policy was in place). Figure 4 clearly shows how in some years the carryover volume is overestimated (year 2013, 2014, and 2016) and in others it is underestimated (year 2012 and year 2015) leading to an overall balance in the evaluated period. The bias in carryover volume for the 2011-2016 period is close to zero given a constant factor value of 78%. It is important to underline that the decisions of farmers to use less water than they are allocated within their entitlement is complex and depends on a range of factors that include not only water availability, but also economic factors and personal preferences. We agree that it would be interesting to extend this study by explicitly including the decisions of farmers and how this feeds back to the basin level allocation decisions. This goes beyond traditional optimisation of the threshold, as it would require for example developing agent-based models to incorporate farmers behaviour. We propose to extend the discussion in section 4.2 (also in response to the second reviewer) by adding the following paragraph:

Allocation decisions made depend not only on the available water in reservoirs and the expected inflows, but also on the actual demand from the crops planted by farmers. In our study demand is taken as the sum of the entitlements of farmers, reduced by the use reduction factor we introduce. Given the water allocated to meet their entitlement, farmers will make their decisions on the crops they plant for the season. In the Murrumbidgee basin, farmers may, however, also trade the water they are entitled to;

or store part of their allocation for use in the next season by deciding to leave it in the upstream reservoirs as carry-over (Horne, 2016). As a result, there are quite complex feedbacks as the decision to carry water allocated over to the next season will influence the allocation decisions at the basin level in that next season. Decisions made by the farmers on what and how much to crop are complex and depend on a range of factors that include the available water through allocation, but also economic factors and personal preferences. The allocation-use reduction factor we introduce to consider these decisions made by farmers, and we find a value an average use of 78% of water entitled to best emulate actual decisions made, on average. While this factor could be optimised mathematically, a detailed understanding of how farmers make decisions is then required. Línes et al (2018) develop a decision model based on interviews of farmers in the Ebro basin in Spain, showing that decisions of what to crop depends on their perception of water availability and will differ between seasons considered wet and seasons considered dry, as well as their aversion to risk and technological capacities. They find that the availability of information on available water as the season develops, such as provided through a seasonal forecast will influence perceptions of water availability and consequently cropping decisions. Further research into how farmers in the Murrumbidgee basin make decisions using for example agent-based models (Wens et al., 2019) could shed more light on the influence on water allocations decisions made at the basin levels.

Wens M., Johnson JM, Zagaria C. Veldkamp T. 2019. Integrating human behavior dynamics into drought risk assessment—A sociohydrologic, agent‐based approach. WIREs Water. https://doi.org/10.1002/wat2.1345

4.    Fourthly, stochastic optimization models are usually set up for reservoir operations using streamflow forecasts.    For some examples, please refer to Labadie (2004,    https://doi.org/10.1061/(ASCE)0733-9496(2004)130:2(93)),Celeste and    Billib    (2009,https://doi.org/10.1016/j.advwatres.2009.06.008),Zhaoet al.(2012,    https://doi.org/10.1029/2011WR010623),    Turner    et

al.(2017,https://doi.org/10.5194/hess-21-4841-2017),and Anghileri-etal.(2019,https://doi.org/10.1029/2019WR025280). One remarkable advantage of stochastic optimization models is the explicit handling of forecast uncertainty. Also, the system performance can be optimized, instead of being simulated. For the case study, is it possible to set up an optimization model?

Reply: We thank the reviewer for this interesting comment. As indicated, and also demonstrated in the work cited and in the review of Labadie (2004), stochastic optimization can be applied in optimising water allocation with explicit consideration of uncertainty. To develop such a stochastic optimisation model would require establishing a (complex) objective function for water allocation and water utilisation in the basin. The results of such a study would be very useful to review the current water allocation policy and regulations. However, we believe that this is beyond the scope of our work as the optimisation of the policy itself is not the priority of our simulations. The focus of our research is to explore how water allocations could benefit through informing the existing policy and regulations using seasonal forecasts, rather than changing or optimising the policy. We believe this is important. The existing policy and regulations have in our understanding been established through careful consultation with the multiple stakeholders in the basin. It is our belief that research in how accepted and currently operational policies and regulations can benefit from the use of seasonal forecast information can encourage the uptake of seasonal forecasts being adopted in practice.

---

## Author Comment (AC2) · 29 Apr 2020

We thank the reviewer for taking the time to review the manuscript and for the helpful comments and suggestions. Here we provide answers to the specific comments and indications of how we propose to improve the manuscript to address the issues raised by the reviewer.

General comments This manuscript presents the benefit of using an ensemble of seasonal streamflow forecasts in water allocation decisions with an emphasis on those decisions in dry seasons and dry years. This is very important for farmers to choose which crop to plant and to decide on the area to be cropped. And also, the manuscript

described the development of new approaches for the reservoir inflow estimates to replace the fixed inflow with the forecasted inflows, decision model to emulate the feedback loop between simulated reservoir storage and water allocation to irrigated crops, inflow forecasts, etc. The authors have briefly evaluated the approaches and identified effectively, and find that there is a quite much higher inconsistency and lower accuracy in estimating water available for allocation during dry seasons and dry years. This is a good and new insight of present manuscript to enhance our understanding of the water allocation for the farmers. The subject is relevant to the journal, the manuscript is well written and structured.

1. However, at present, the focus of manuscript is not particularly strong and it seems that the authors are not entirely sure about the key message they wish to convey. There are some aspects are suspected as follows: Firstly, the equations (on pages 6-8) to determine the available water for allocation needs more variables related to complicated relationships among the water demands and feedback loop among the reservoirs.

Reply: We appreciate the comment of the reviewer on the clarity of the message we would like to convey. To strengthen that message, we propose to revise the last sentence of the abstract as follows: "Our results show that seasonal streamflow forecasts can provide benefit in informing water allocation policies, particularly through earlier establishing final water allocations to farmers in the irrigation season. This allows them to plan better and use water allocated more efficiently".

We also propose to make the three key messages accompanying the paper clearer:

1. The existing water allocation policy in a highly regulated basin is emulated in a decision model, and subsequently extended to inform allocation decisions with a seasonal streamflow forecast. 2. Using the FoGSS seasonal forecast to inform allocation decisions is shown to allow final annual allocations to farmers to be established one and a half months earlier than under the current policy. This is important as it helps farmers

plan better and use allocated water more efficiently. 3. FoGSS forecasts derived from the POAMA GCM data perform marginally better than those derived from resampled climatology (ESP+); though forecast uncertainty requires a trade-off between better estimates of available water and the cost of downward revisions of water allocations to farmers.

Regarding the equations to determine the available water for allocation, these have been established to include several variables such (in order of priority); water allocated to meet environmental needs; town water allocations; high security allocation; irrigation, conveyance losses, and finally general security allocation, which is allocation that is the focus of this research. Note that the water available for allocation is the total volume in the (two) reservoirs, plus the expected inflow which is derived from the streamflow forecast. While we agree that there quite complex interactions, we have established these equations based on the current water policy and regulations in the basin. In doing so we have purposefully kept the equations as simple as possible while staying true to the policy. Through comparison with recorded allocation decisions made under the existing policy we demonstrate that these decisions are reasonably well emulated.

2. Secondly, it is necessary to discuss the nonlinear processes of higher water demands and tradeoffs among the water users and reservoirs behind dams in the study area in dry seasons and dry years. These processes are suggested to presented more in detail in the context "4.2 To what degree does the seasonal forecast help in the decision process?".

Reply: We agree that the interaction between water demand and availability, and tradoffs made is highly complex. We propose to extend the discussion in section 4.2 to include these interactions. The following paragraph will be added to Section 4.2.

Allocation decisions made depend not only on the available water in reservoirs and the expected inflows, but also on the actual demand from the crops planted by farmers.

In our study demand is taken as the sum of the entitlements of farmers, reduced by the use reduction factor we introduce. Given the water allocated to meet their entitlement, farmers will make their decisions on the crops they plant for the season. In the Murrumbidgee basin, farmers may, however, also trade the water they are entitled to; or store part of their allocation for use in the next season by deciding to leave it in the upstream reservoirs as carry-over (Horne, 2016). As a result, there are quite complex feedbacks as the decision to carry water allocated over to the next season will influence the allocation decisions at the basin level in that next season. Decisions made by the farmers on what and how much to crop are complex and depend on a range of factors that include the available water through allocation, but also economic factors and personal preferences. The allocation-use reduction factor we introduce to consider these decisions made by farmers, and we find a value an average use of 78% of water entitled to best emulate actual decisions made, on average. While this factor could be optimised mathematically, a detailed understanding of how farmers make decisions is then required. Línes et al (2018) develop a decision model based on interviews of farmers in the Ebro basin in Spain, showing that decisions of what to crop depends on their perception of water availability and will differ between seasons considered wet and seasons considered dry, as well as their aversion to risk and technological capacities. They find that the availability of information on available water as the season develops, such as provided through a seasonal forecast will influence perceptions of water availability and consequently cropping decisions. Further research into how farmers in the Murrumbidgee basin make decisions using for example agent-based models (Wens et al., 2019) could shed more light on the influence on water allocations decisions made at the basin levels.

Wens M., Johnson JM, Zagaria C. Veldkamp T. 2019. Integrating human behavior dynamics into drought risk assessment—A sociohydrologic, agent‐based approach. WIREs Water. https://doi.org/10.1002/wat2.1345

3. Thirdly, the better quality of figures in the text and supplementary materials are
suggested to provide. Reply: We will make sure quality of the figures will be improved and will include these to a higher resolution.
* * *